# Morphology Effects on Structure-Activity Relationship of Pd/Y-ZrO$_2$ Catalysts for Methane Oxidation

**Xiujuan Zhang [1], Tingting Zheng [1,2,\*], Jiangli Ma [1,2], Chengxiong Wang [1,2], Dongxia Yang [1,2,3,\*] and Ping Ning [3]**

[1] State Key Laboratory of Advanced Technologies for Comprehensive Utilization of Platinum Metals, Kunming Institute of Precious Metals, Kunming 650106, China

[2] State-Local Joint Engineering Laboratory of Precious Metal Catalytic Technology and Application, Kunming Sino-Platinum Metals Catalysts Co., Ltd., Kunming 650106, China

[3] Faculty of Environmental Science and Engineering, Kunming University of Science and Technology, Kunming 650093, China

\* Correspondence: tingting.zheng@spmcatalyst.com (T.Z.); doris.yang@spmcatalyst.com (D.Y.)

**Abstract:** Pd/Y-ZrO$_2$ catalysts were prepared by Y-ZrO$_2$ with different morphologies (flower-like, spherical, reticulated, and bulk-specific morphology), which were prepared by hydrothermal synthesis. Activity eValuation and characterization results show that the morphology influences the microstructures of Y-ZrO$_2$ and the chemical states of active Pd species, thus affecting the activity of methane oxidation. Bulk Pd/Y-ZrO$_2$ exhibits the best CH$_4$ oxidation activity and thermal stability due to the block shape exposed (101) surface, and the single tetragonal phase structure maintained after high-temperature aging. The relatively large-sized Pd particles and Pd$^0$ jointly promote the catalytic oxidation of CH$_4$.

**Keywords:** Y-ZrO$_2$; Pd/Y-ZrO$_2$; morphology effect; methane oxidation; structure-activity relationship

## 1. Introduction

The support has a significant influence on metal dispersion, chemical states, catalyst activity, and stability. In the field of natural gas vehicle exhaust purification, one of the most effective catalysts for methane oxidation is Pd/γ-Al$_2$O$_3$. However, it becomes unstable at high temperatures and deactivates when exposed to water. In order to solve those problems, many support materials have been studied, including SiO$_2$-Al$_2$O$_3$ [1], SnO$_2$ [2], rare earth (La, Ce) [3,4], and transition metal (Zr, Nb) [5,6] doped Al$_2$O$_3$, CeO$_2$-ZrO$_2$ [7], AB$_2$O$_4$ spinel-type [8] and ABO$_3$ perovskite-type oxides et al. [9,10]. Previous studies have shown that ZrO$_2$ is prone to generate oxygen vacancy due to its redox properties and reported as more active and stable than that alumina in methane oxidation [11–13]. However, ZrO$_2$ has a low surface area and tends to sinter under high temperature. Some researchers have found new methods to prepare ZrO$_2$ with a high surface area to improve its activity [14]. In addition, other studies have also shown that rare earth modification can stabilize the structure of CeO$_2$–ZrO$_2$, which has been widely used in three-way catalysts due to the excellent thermal stability and light-off activity in Pd/Rh catalysts [15]. Our previous studies found that rare earth oxide doping could further improve the performance of ZrO$_2$. Both La and Y had an obvious effect on improving the thermal stability of ZrO$_2$, in which Y doping revealed the best catalytic activity for CH$_4$ oxidation [16].

In addition, the morphology and structure of the support also have a great influence on the performance of the catalyst. The morphology, structure, crystal plane effect, and interaction between supports can be controlled by preparation methods. In recent years, to adjust the structure and catalytic activity of the catalyst, a large number of studies have been carried out on the structure and morphology of zirconium-based catalysts in automobile exhaust catalytic reactions [17–24]. Ren et al. [25] have reported that the samples prepared by the sol-gel method using citric acid exhibited more homogeneous particle

sizes and higher specific surface areas than that of the urea grind combustion method, which is beneficial for oxygen release. Jie [26] synthesized four $CeO_2$-$ZrO_2$ oxides with different morphologies by hydrothermal synthesis (polyhedra, rods, and plates) and the coprecipitation method (disordered), respectively. Polyhedral $CeO_2$-$ZrO_2$ can promote the migration of bulk oxygen species to the surface and obtain better performance due to the increase in micro lattice strain caused by lattice distortion and defects. Disordered $Rh/CeO_2$-$ZrO_2$ catalyst had a high specific surface area and wide static oxygen storage capacity (OSC) operation window. Jie and his team [27] recently reported that $Pd/CeO_2$-$ZrO_2$ catalysts with controlled morphologies had different exposed crystal planes. The TWC performance of $Pd/CeO_2$–$ZrO_2$ presents a clear "structure-activity" relationship towards $CeO_2$-$ZrO_2$ morphologies.

On the basis of our previous research [16], nano Y-$ZrO_2$ materials with special structure, morphology, and exposed crystal surface were synthesized by hydrothermal synthesis to achieve the controllable Y-modified $Pd/Y$-$ZrO_2$ catalyst with high specific surface area, high catalytic activity, and stable active Pd species. Meanwhile, the structure-activity relationship between material structure and properties also be studied.

## 2. Experimental

### 2.1. Catalysts Preparation

Flower-like Y-$ZrO_2$ composite oxide with yttrium oxide content of 10 wt% was prepared by a previously reported hydrothermal synthesis [28]. 4.44 g $Zr(SO_4)_2 \cdot 4H_2O$(AR), 0.36 g $Y(NO_3)_3 \cdot 6H_2O$(AR), and 0.58 g NaOH (AR) was dissolved into 150 mL deionized water and stirred for 60 min. The mixed solution was then subjected to hydrothermal treatment in a 200 mL Teflon-lined stainless-steel autoclave at 200 °C for 6 h. The obtained products were filtered and washed with an equal volume of deionized water 4 times, followed by drying at 100 °C overnight and then calcined in air at 800 °C for 4 h. The resulting sample is marked as FlYZr.

Spherical Y-$ZrO_2$ composite oxide with yttrium oxide content of 10 wt% was prepared by a previously reported solvothermal method [29]. 2.42 g $ZrOCl_2 \cdot 8H_2O$ (AR) and 0.35 g $Y(NO_3)_3 \cdot 6H_2O$(AR) were dissolved in a 150mL mixture acetylacetone and n-butanol (the volume ratio is 1:1), then stirred until totally transparent. The mixture was transferred to a 200 mL Teflon-lined stainless and heated at 65 °C for 4 h, then reacted at 200 °C for 12 h. The obtained products were filtered and washed with an equal volume of absolute ethanol 2 times and deionized water 2 times, followed by drying at 75 °C overnight and then calcined in air at 550 °C for 4 h. The resulting sample is marked as SpYZr.

Reticular Y-$ZrO_2$ composite oxide with yttrium oxide content of 10 wt% was prepared by EDTA-assisted hydrothermal synthesis. 14.16 g $ZrOCl_2 \cdot 8H_2O$(AR), 2.04 g $Y(NO_3)_3 \cdot 6H_2O$(AR), and 1.97 g NaOH(AR) were dissolved into 150 mL ethanol and stirred for 60 min. Then 7.2 g EDTA was added to the mixed solution above and stirred for 60 min. The mixed solution was then subjected to hydrothermal treatment in a 200 mL stainless-steel autoclave at 120 °C for 4 h. The obtained products were filtered and washed with an equal volume of deionized water 4 times, followed by drying at 100 °C overnight and then calcined in air at 800 °C for 4 h. The resulting sample is marked as ReYZr.

Bulk-shaped Y-$ZrO_2$ composite oxide with yttrium oxide content of 10 wt% was prepared by a CTAB-assisted hydrothermal synthesis. Zirconium and yttrium were precipitated successively by co-precipitation. Dropped $ZrO(NO_3)_2$(AR) solution(16.89 g $ZrO(NO_3)_2$ dissolved in 90 mL deionized water) and NaOH(AR) solution(10 g NaOH dissolved in 50 mL deionized water) into the beaker at the same time and stirred, while maintaining pH equal to 7. Then, dropped $Y(NO_3)_3$ solution(3.34 g $Y(NO_3)_3 \cdot 6H_2O$ dissolved in 10 mL deionized water) and NaOH solution into the mixture above and kept string while maintaining pH equal to 9. Then 0.5 g CTAB was added and stirred for 60 min. The mixture was then subjected to hydrothermal treatment in a 200 mL stainless-steel autoclave at 160 °C for 4 h. The obtained products were filtered and washed with an

equal volume of deionized water 4 times, followed by drying at 100 °C overnight and then calcined in air at 550 °C for 4 h. The resulting sample is marked as BuYZr.

The corresponding supported Pd/Y-ZrO$_2$ catalysts (Pd content was 0.5%) were prepared by a standard conventional impregnation method using an aqueous of [Pd(NH$_3$)$_4$](NO$_3$)$_2$ as the metal precursor. An amount of [Pd(NH$_3$)$_4$](NO$_3$)$_2$ was weighed into de-ionized water and stirred until dissolved. Then the Y-ZrO$_2$ support was added to the solution and stood for 24 h at room temperature. The impregnated samples were dried in an oven at 100 °C overnight and then calcined in a muffle furnace at 550 °C for 3 h. The prepared catalysts were recorded as Pd/FlYZr, Pd/SpYZr, Pd/BuYZr, and Pd/ReYZr, respectively. The corresponding aged Pd/Y-ZrO$_2$ catalyst was obtained by calcination of the fresh catalyst in a muffle furnace at 950 °C for 4 h.

### 2.2. Catalytic Activity Tests

The methane oxidation activity of the catalysts was carried out on the multifunctional micro-fixed-bed reactor. The gas composition was analyzed online using the Fourier infrared (FTIR) gas analyzer of (MKS). All the catalysts were pellet-pressed and sieved to 40~60 mesh. 0.5 g of 40~60 mesh catalysts was placed into a quartz reaction tube with an inner diameter of 8 mm. The reaction mixture with the stoichiometric ratio consisted of 2000 ppm CH$_4$, 3500 ppm CO, 0.525% O$_2$, 12% CO$_2$, 1000 ppm NO, 8% H$_2$O, and N$_2$ as balance gas. The mixed reaction gas was introduced into the reactor at a GHSV of 80,000 h$^{-1}$ until it was balanced at 120 °C, then heated to 600 °C with a rate of 10 °C/min for pre-treatment, followed by cooling to 100 °C. The activity test was accomplished by the same atmosphere and temperature range as the pre-treatment did.

### 2.3. Catalysts Characterizations

The morphology of samples was performed using a Nova nanosem 450 field emission scanning electron microscopes (SEM) with high stability schottky field emission electron gun, and a magnification range was 35~1,000,000 times.

The textural properties of samples were measured by nitrogen physisorption at about −196 °C using a Quantachrome NOVA2000e analyzer. Before analysis, the powder samples were first degassed at 200 °C under a vacuum condition for 15min, then 100 mg degassed samples were used for measurements. The specific surface area was eValuated from the adsorption data in the P/P0 range of 0.05–0.2 using the Brunauer-Emmett-Teller (BET) equation. The pore size distributions were analyzed by MIP using a Micromeritics Poresizer 9320. The total pore volumes were calculated from the single nitrogen absorption amount at the P/P$_0$ of about 0.97.

Powder X-ray diffraction (XRD) analysis was performed on a D/MAX-2200 diffractometer with Cu K$\alpha$ radiation ($\lambda$ = 0.15418 nm) operated at 40 kV and 40 mA. The crystalline size was calculated according to the Scherrer equation:

$$D = K\lambda/\beta cos\theta$$

where, *K* was the Scherrer constant, which was equal to 0.89, generally 1. *D* represented the grain size (nm), $\beta$ represented the integral half-height width, which shall be converted into radians (rad) during calculation, $\theta$ was the diffraction angle, and $\lambda$ was the wavelength of the X-ray.

The dispersion of Pd was quantified by using CO pulse adsorption methods on the CHEMBET 3000 analyzer. 0.1 g of the sample was heated in H$_2$ (10%)/Ar (75 mL·min$^{-1}$) from room temperature to 450 °C and held for 1 h. After a subsequent purge in He flow, the sample was cooled to 50 °C, then injected same doses of CO per time until no changes in the TCD signal were found. Then calculated, the dispersion (*D*) of Pd was according to the adsorption amount of CO (Equation (1)). The mean Pd particle diameters

$d$(nm) was estimated from dispersion data using the following Equation (2) derived from hemispherical particles.

$$D(\%) = \frac{V_{ads} \times K \times M}{V_m \times W_s \times C} \times 100\% \tag{1}$$

$$d(nm) = \frac{6 \times 10^9 \times A \times M}{\rho D n} \tag{2}$$

Here, $V_{ads}$ was the CO adsorption volume (cm$^3$/g), and $K$ was the correction factor which was equal to 1 in this experiment. $M$ represented the atomic mass of Pd (106.42 g/mol), $V_m$ represented the molar volume of adsorbed gas (22,400 cm$^3$/g), $W_s$ represented the sample mass (g), and $C$ represented the percentage content of Pd(%). $A$ represents the surface concentration of Pd atoms, which was equal to $1.27 \times 10^{19}$, $\rho$ was the volumetric mass of Pd, which was equal to $12.02 \times 10^6$ g/m$^3$, $D$ was palladium dispersion, and $n$ was Avogadro number of $6.02 \times 10^{23}$.

O$_2$ temperature programmed oxidation (O$_2$-TPO) analysis was completed on CHEM-BET 3000 analyzer with a thermal conductivity detector (TCD). 0.1 g catalyst was heated from room temperature to 450 °C at a rate of 10 °C·min$^{-1}$ in The atmosphere, reduced to 200 °C after constant temperature pretreatment for 1h. After then, the inlet gas was switched to 10% O$_2$/N$_2$. The O$_2$-TPO profile was collected by simultaneously raising the reactor temperature from room temperature to 900 °C at 10 °C/min and cooling it to room temperature at the same speed. The whole process gas flow was 75 mL/min.

High-resolution transmission electron microscopy (HRTEM) observation was performed using a JEM-2100 at 200 kV. The samples were firstly dispersed in ethanol and then deposited onto a copper grid-supported carbon film.

X-ray photoelectron spectra (XPS) were recorded on a Thermo K-Alpha system employed with monochromated Al K$\alpha$(1486.6 eV) radiation operating at 72 W. The binding energies were calibrated using the C1s peak of contaminant carbon (B.E. = 284.8 eV) as standard.

## 3. Result and Discussion

### 3.1. Physicochemical Properties of Different Y-ZrO$_2$ Mixed Oxides

The morphologies of the obtained Y-ZrO$_2$ samples were observed by SEM. As shown in Figure 1, the fresh FlYZr sample is a flower-like structure with a uniform size (about 1μm) distribution and extremely thin sheet-like nanostructures. Fresh SpYZr sample is a hollow spherical structure with a size of about 1μm and a thickness of about 80 nm. Fresh ReYZr sample is composed of worm-like nanoparticles with a length of about 60 nm stacked on each other to form a multilayer network structure. BuYZr sample shows an amorphous bulk structure composed of ellipsoidal nanoparticles of about 10 nm stacked randomly.

Figure 2 shows the SEM images of aged Y-ZrO$_2$ mixed oxides with different morphologies. It can be observed that the FLYZr sample still maintains a flower-like structure that consists of sheet-like nanostructures after aging. The SpYZr sample shows a hollow spherical structure with obvious sintering. Part of the hollow spherical structure was destroyed, and a sheet-like structure formed by the arrangement of spherical nanoparticles appeared. The aged ReYZr samples show a clearer multilayer network structure with obvious growth. The ellipsoidal nano-particles of the aged BuYZr sample were growing up, but they were still disorderly piled up into lumps.

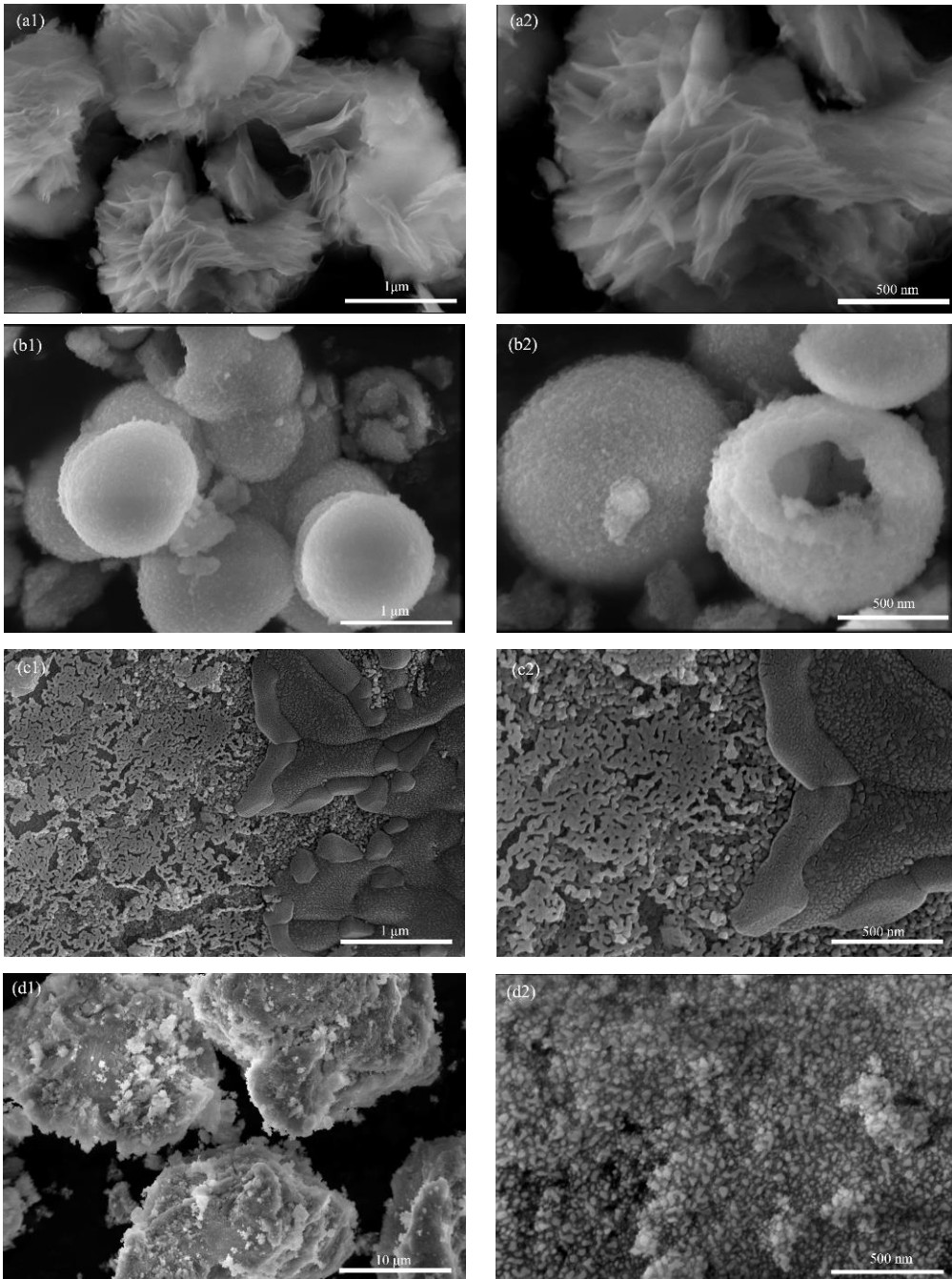

**Figure 1.** SEM images of fresh FlYZr (**a1**,**a2**), SpYZr (**b1**,**b2**), ReYZr (**c1**,**c2**), BuYZr (**d1**,**d2**).

Figure 3 shows the obtained isotherms, and Table 1 summarizes the surface area, pore volume, and average pore diameter of all the samples. Among the fresh samples, SpYZr and BuYZr samples have larger specific surface areas, 82.7 $m^2 \cdot g^{-1}$, and 106.7 $m^2 \cdot g^{-1}$, respectively. A higher surface area of support favors Pd dispersion and sintering resistance [30]. The ReYZr sample has the smallest specific surface area, which is only 6.5 $m^2 \cdot g^{-1}$. The specific surface area of the material drops sharply, the pore volume is obviously reduced, and the average pore diameter increases under high temperatures. Among them, the FlYZr and BuYZr samples still have a relatively large specific surface area.

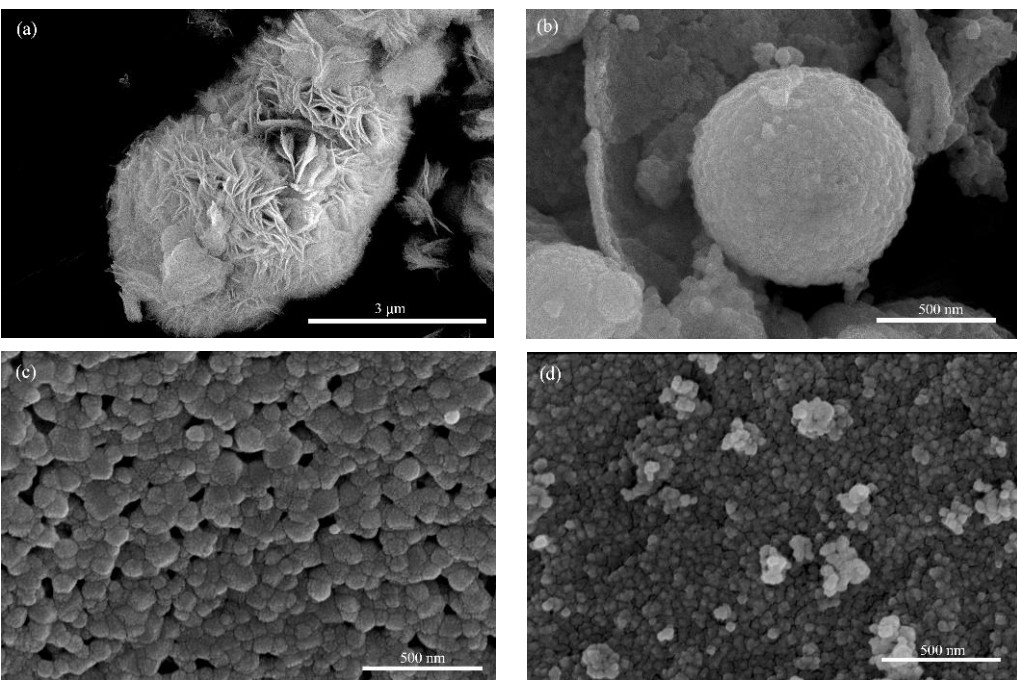

**Figure 2.** SEM images of Aged FlYZr (**a**), SpYZr (**b**), ReYZr (**c**), and BuYZr (**d**).

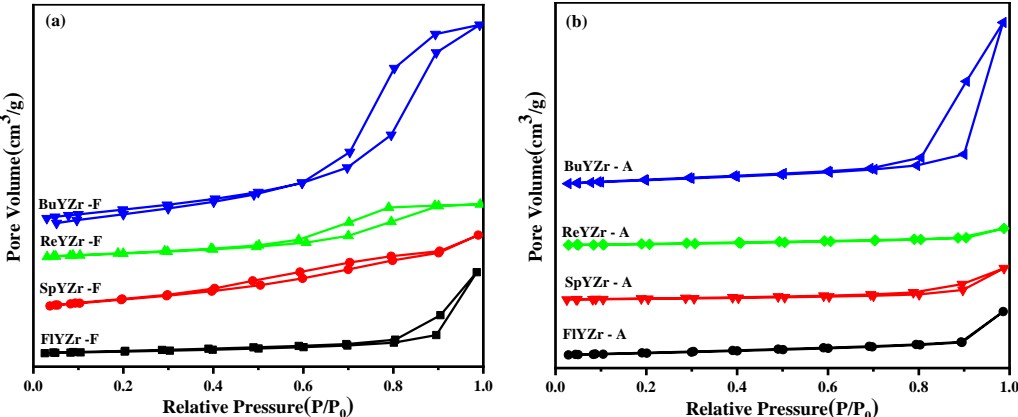

**Figure 3.** Adsorption-desorption isotherms of fresh (**a**) and aged (**b**)Y-ZrO$_2$ compositbe oxides with different morphologies.

**Table 1.** Specific surface area and pore structure parameters of Y-ZrO$_2$ composite oxides with different morphologies.

| Samples | $S_{BET}$ (m$^2 \cdot$g$^{-1}$) | | $V_P$ (cm$^3 \cdot$g$^{-1}$) | | $D_P$ (nm) | |
|---|---|---|---|---|---|---|
| | Fresh | Aged | Fresh | Aged | Fresh | Aged |
| FlYZr | 22.0 | 14.2 | 0.142 | 0.063 | 12.9 | 9.0 |
| SpYZr | 82.7 | 8.8 | 0.142 | 0.056 | 3.4 | 10.6 |
| ReYZr | 6.5 | 6.9 | 0.022 | 0.025 | 5.6 | 7.2 |
| BuYZr | 106.7 | 34.9 | 0.353 | 0.235 | 6.6 | 13.5 |

As shown in Figure 3a, the fresh SpYZr, ReYZr, and BuYZr samples have similar adsorption isotherms and hysteresis loops. The isotherms were categorized as IV isotherms with an approximate H$_2$-type hysteresis loop characteristic of cylindrical pores [31,32], and the sample hole type is a cylindrical ink bottle mixed channel. The characteristic of this channel is that the size of the orifice is similar to that of the channel/cavity, which is

conducive to the mass and heat transfer in the reaction. The network sample has an obvious saturated adsorption platform, indicating that its pore structure is relatively uniform [33]. After high-temperature aging, all the Y-ZrO$_2$ samples show an H3 hysteresis ring with a hole type of slit-type channel, which is formed by sintering and mutual accumulation of particles. It can be seen that the Y-ZrO$_2$ sample has obvious sintering and agglomeration after aging [34]. Among them, the hysteresis loop of the flower-like and net-like samples is not obvious, indicating destruction of the mesoporous pores. The BuYZr sample owns the most mesoporous structure after aging, which proves that it has good thermal stability.

The phase states of supports have a great influence on the performance of supported Pd catalysts [35–37]. Figure 4 presents the XRD patterns of all the obtained Y-ZrO$_2$ with different morphologies. The peaks at 24.1°, 28.2°, 31.5° are exclusively indexed to (011), (−111) and (111) planes of monoclinic ZrO$_2$ phase(JCPDS # 86-1449). Diffraction peaks appeared around 30.1°, 34.9°, 50.1°, 59.6°, 62.6°, and 73.6°can be assigned to the (101), (110), (112), (202), (220) crystal planes of tetragonal Y$_x$Z$_{1-x}$O$_2$ (JCPDS # 82-1244). As shown in Figure 4a,c, the fresh flower-like, and reticular Y-ZrO$_2$ show a mixed phase structure with a small amount of monoclinic phase, and the proportion of monoclinic phase increases after aging. In comparison, the spherical and bulk-shaped Y-ZrO$_2$ sample presents a single tetragonal phase, no matter whether it is fresh or aged. In addition, the diffraction peaks are enhanced after aging. It is indicated that the spherical and bulk-shaped Y-ZrO$_2$ sample has good thermal stability.

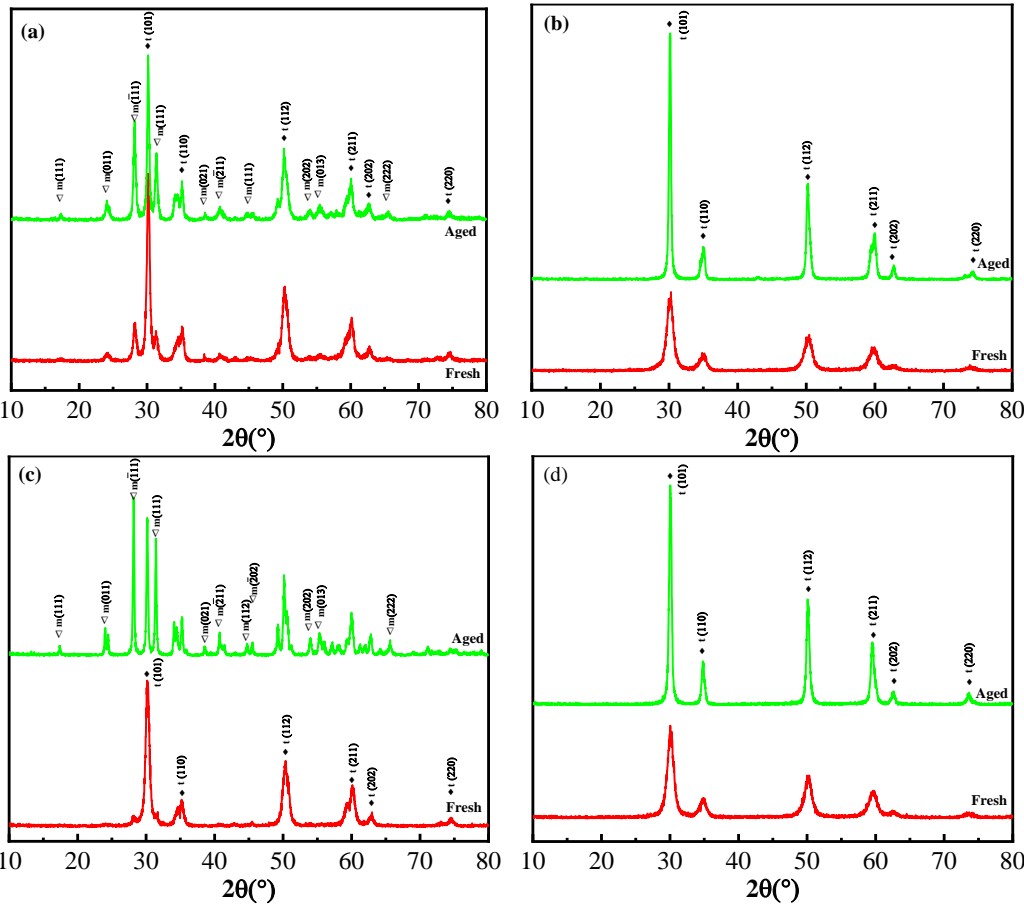

**Figure 4.** XRD patterns of FlYZr (**a**), SpYZr (**b**), ReYZr (**c**), and BuYZr (**d**), black rhomboid (tetragonal), white triangle (monoclinic).

Figure 5 shows the crystalline size of Y-ZrO$_2$ oxides with different morphologies. It can be seen that both the fresh and aged bulk. Y-ZrO$_2$ samples have a smaller particle size,

indicating that they have good anti-sintering ability, which is consistent with the results of SEM analysis.

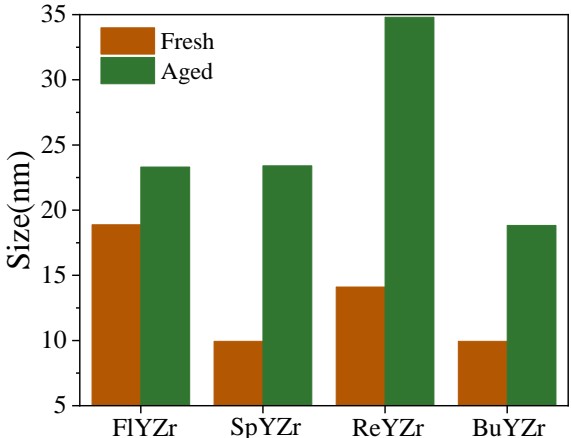

**Figure 5.** Statistical chart of average particle size of Y-ZrO$_2$ composite oxides with different morphologies.

### 3.2. Structural and Textural Properties of Pd/Y-ZrO$_2$ Catalysts with Different Morphologies

The methane oxidation reaction on the supported Pd catalyst is structurally sensitive, which attribute to the positive effect of Pd particle size on the performance [37,38]. Since Y, Zr, and Pd have similar contrasts, the dispersion and particle size of Pd are measured by CO chemisorption measurements. As shown in Table 2, the Pd dispersion of the fresh catalyst is positively correlated with the specific surface area of the Y-ZrO$_2$ support due to enough surface for Pd loadings. Pd/SpYZr and Pd/BuYZr catalysts have high Pd dispersion and thermal stability, which are related to the more stable tetragonal phase structure of SpYZr and BuYZr oxides.

**Table 2.** Pd dispersion and particle size of Pd/Y-ZrO2 catalysts with different morphologies.

| Samples | Pd Content (%) | Pd Dispersion (%) | | Pd Particle Size (nm) | |
|---|---|---|---|---|---|
| | Fresh | Fresh | Aged | Fresh | Aged |
| Pd/FlYZr | 0.41 | 17.6 | 2.9 | 6.4 | 38.3 |
| Pd/SpYZr | 0.46 | 49.4 | 25.9 | 2.3 | 4.3 |
| Pd/ReYZr | 0.42 | 10.5 | 3.4 | 10.6 | 33.1 |
| Pd/BuYZr | 0.41 | 57.0 | 25.7 | 2.0 | 4.4 |

The morphologies of the obtained Pd/Y-ZrO$_2$ are observed by TEM and HRTEM. As shown in Figure 6, obvious Pd particles can be seen from the HRTEM image of Pd/FlYZr. Its particle size is within 5–10 nm, which is basically consistent with the calculated results of dispersion (Table 2). The HRTEM image of Pd/FlYZr (a2) shows that the exposed crystal plane of ZrO$_2$ is mainly the (101) plane of t-ZrO$_2$. TEM images of Pd/SpYZr, Pd/ReYZr, and Pd/BuYZr (b1, c1, d1) show Y-ZrO$_2$ particles with a size of about 10 nm, 15 nm, and 10 nm. HRTEM (b2, c2, d2) image shows that the exposed crystal plane of SpYZr is mainly attributed to the (101) plane of t-Y$_x$Z$_{1-x}$O$_2$. The exposed crystal planes of ReYZr are the (011), (−111), and (111) planes of m-Y$_x$Z$_{1-x}$O$_2$, of which the (−111) plane is the main one. The exposed crystal plane of BuYZr is the (101) plane of t-Y$_x$Z$_{1-x}$O$_2$. Among the exposed plane, the (−111) plane is the most stable surface of monoclinic zirconia [39], and the (101) plane of t-ZrO$_2$ is more prone to distortion. This lattice strain or volume defect can promote the migration of oxygen from the body to the surface and improve activity. Therefore, the (101) surface of t-ZrO$_2$ is more reactive than the (−111) surface of m-ZrO$_2$ [26].

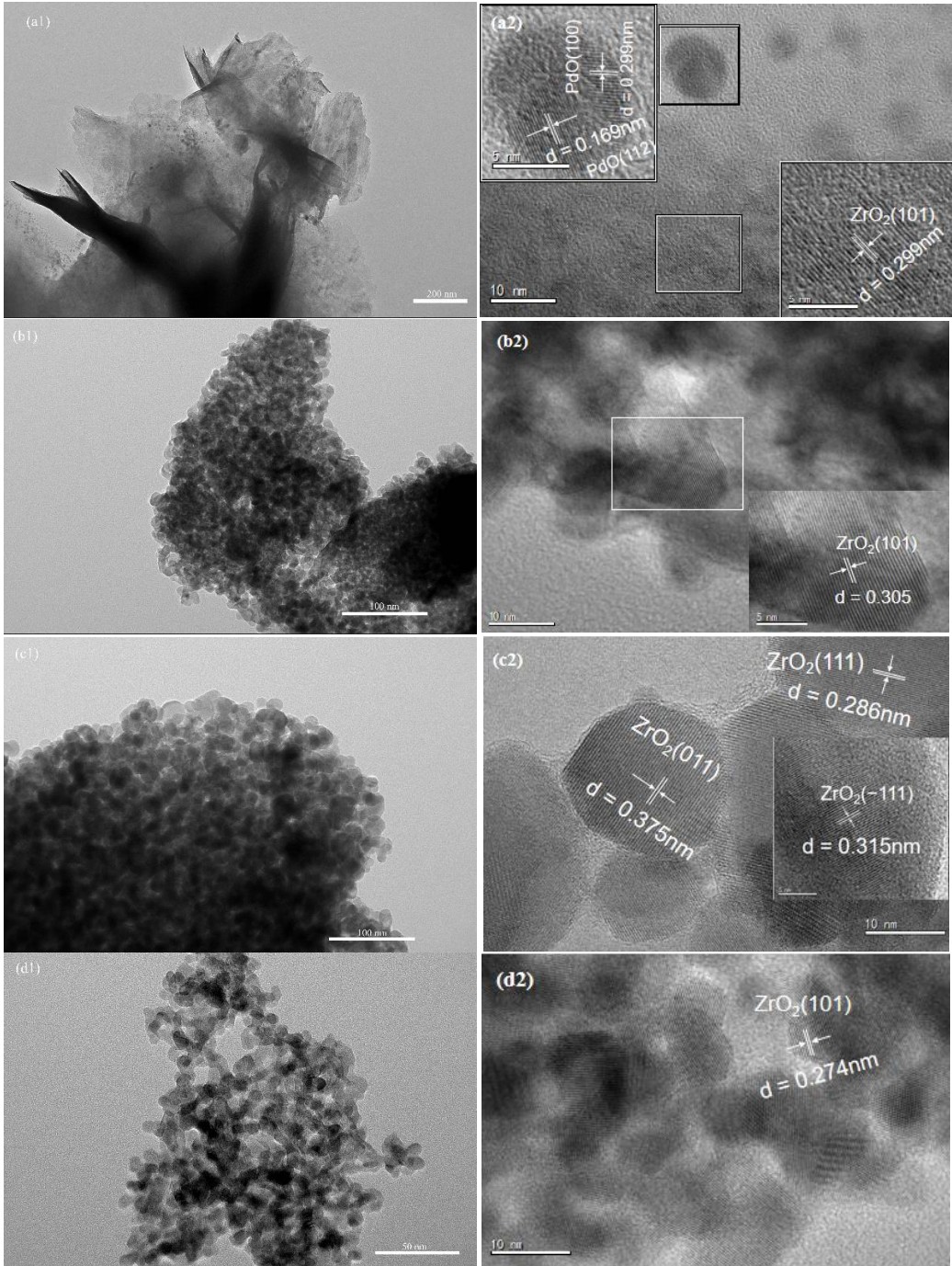

**Figure 6.** TEM images of Pd/FlYZr (**a1**), Pd/SpYZr (**b1**), Pd/ReYZr (**c1**), Pd/BuYZr (**d1**), and HRTEM images for Pd/FlYZr (**a2**), Pd/SpYZr (**b2**), Pd/ReYZr (**c2**), Pd/BuYZr (**d2**).

Redox properties of palladium-supported samples with different morphologies are characterized by TPO experiments. As shown in Figure 7, catalysts with different morphologies differ in the temperatures of PdO decomposition (or thermal reduction) during heating and redox during cooling. The decomposition temperature of PdO on the Pd/SpYZr catalyst is 55~80 °C higher than other fresh samples, which indicates that the PdO species on Pd/SpYZr surface have high thermal stability. The Pd/BuYZr catalyst has larger peak widths of the PdO decomposition peak during heating and redox peak during cooling, indicating that PdO on the Pd/BuYZr catalyst has good dispersibility. It is consistent with the results of the CO chemical adsorption analysis. After aging, the Pd/SpYZr catalyst

shows two PdO$_x$ decomposition peaks during heating, which may correspond to PdO$_x$ species with different valence states.

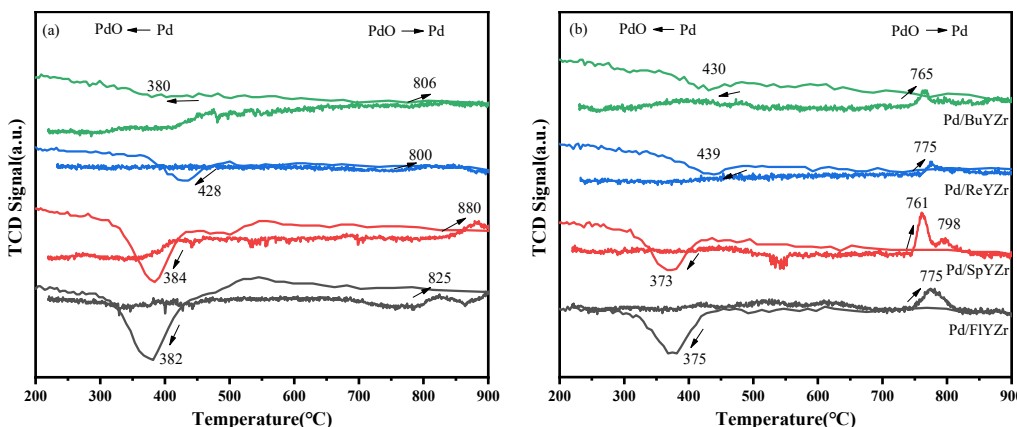

**Figure 7.** O$_2$-TPO spectra of fresh (**a**) and aged (**b**) Pd/Y-ZrO$_2$ catalysts with different morphologies.

In order to further understand the chemical state of palladium species on the surface of Pd/Y-ZrO$_2$ with different morphologies, XPS measurement was carried out, and the results are shown in Figure 8. There are six electron binding energy peaks in the Pd 3d XPS spectrum of the Pd/Y-ZrO$_2$. Two signals around 336.6 eV and 342.4 eV are attributed to the characteristic peaks of Pd$^{2+}$ 3d$_{5/2}$ and Pd$^{2+}$ 3d$_{3/2}$, respectively. The other respective signals of 335.5 eV and 341.6 eV are attributed to the characteristic peaks of Pd$^0$ 3d$_{5/2}$ and Pd$^0$ 3d$_{3/2}$, respectively. In addition, the peaks with bond energies around 332.6 eV and 346.3 eV are attributed to the characteristic peaks of Zr 3p$_{3/2}$ and Zr 3p$_{1/2}$, respectively [40]. The relative percentage of Pd species can be obtained by the area ratio of the Pd$^{2+}$ 3d$_{5/2}$ and Pd$^0$ 3d$_{5/2}$ peaks, as shown in Table 3. It can be seen that the Pd$^{2+}$ content of the flower-like, spherical, and bulk catalysts are relatively low, indicating that the active metal Pd in the Pd/FlYZr, Pd/SpYZr, and Pd/BuYZr catalysts are hard to be oxidized [41]. After aging, the relative percentage of Pd$^{2+}$ on the surface of Pd/FlYZr, Pd/SpYZr, Pd/ReYZr, and Pd/BuYZr catalysts decreased significantly due to the PdO decomposed into metallic Pd$^0$ under high temperature. The interaction between Pd and Y$_x$Zr$_{1-x}$O$_2$ is weakened, thereby promoting the decomposition of PdO.

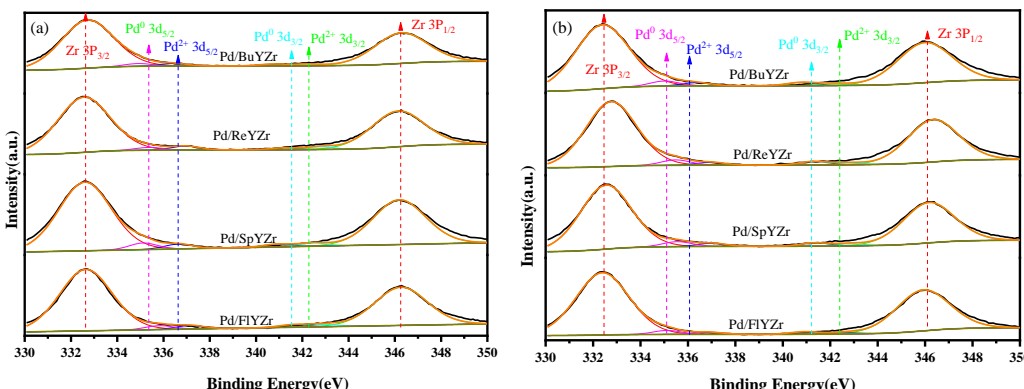

**Figure 8.** Pd 3d XPS spectra of fresh (**a**) and aged (**b**) Pd/Y-ZrO$_2$ catalysts with different morphologies.

**Table 3.** Pd 3d XPS data of Pd/Y-ZrO$_2$ catalysts with different morphologies.

| Samples | Pd 3d$_{5/2}$ Binding Energy (eV) | | Pd$^{2+}$/(Pd$^0$ + Pd$^{2+}$) (%) |
|---|---|---|---|
| | Pd$^0$ | Pd$^{2+}$ | |
| Pd/FlYZr-F | 353.82 | 336.94 | 47.27% |
| Pd/SpYZr-F | 335.12 | 336.64 | 45.87% |
| Pd/ReYZr-F | 335.37 | 336.94 | 58.65% |
| Pd/BuYZr-F | 334.91 | 336.39 | 47.85% |
| Pd/FlYZr-A | 335.13 | 336.82 | 36.79% |
| Pd/SpYZr-A | 335.54 | 337.01 | 38.73% |
| Pd/ReYZr-A | 335.48 | 336.96 | 34.16% |
| Pd/BuYZr-A | 335.00 | 336.34 | 33.94% |

*3.3. Catalytic Performance*

The catalytic activity of CH$_4$ oxidation on Pd/Y-ZrO$_2$ catalysts with different morphologies is tested, and the corresponding results are shown in Figure 9 and Table 4 as a function of temperature range from 350 to 550 °C. The CH$_4$ oxidation activity sequence of fresh Pd/Y-ZrO$_2$ catalyst is followed by Pd/BuYZr > Pd/FlYZr > Pd/SpYZr > Pd/ReYZr without gaseous water. It is apparent that the activity is eVidently inhibited when 10 vol % of water is added due to the blocking by both water and hydroxyl species. This observation is in agreement with most studies in the literature [42–44]. It is interesting that the Pd/SpYZr catalyst showed higher methane oxidation activity than Pd/BuYZr in moisture. Some research groups have shown that water inhibits the combustion reaction when the catalyst is in the oxide phase. However, it does not affect the catalytic activity of metal particles [42]. As shown in Table 3, the relative percentage of Pd$^{2+}$ on the surface of Pd/SpYZr is lower than Pd/BuYZ, resulting in the inhibitory effect of water on Pd/BuYZ greater than Pd/SpYZr. In addition, the large Pd particle size is supposed to contribute better performance over Pd/SpYZr catalyst.

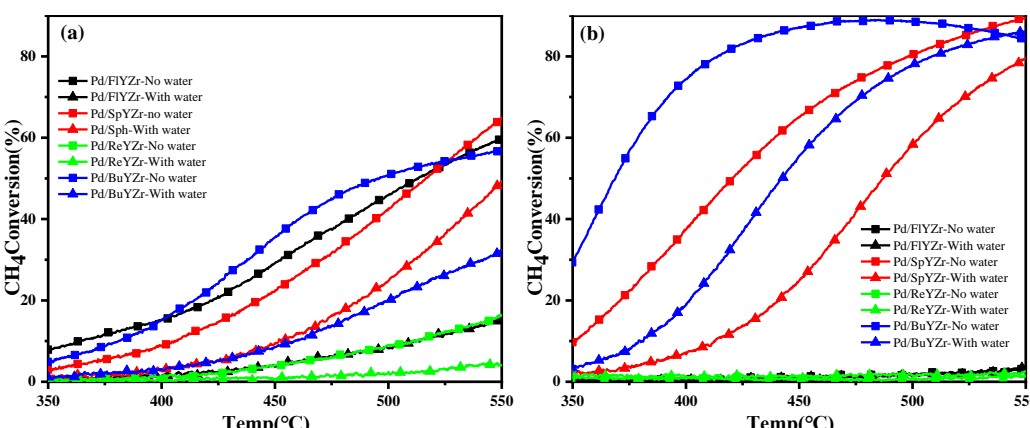

**Figure 9.** CH$_4$ conversion curves of fresh (**a**) and aged (**b**) Pd/Y-ZrO$_2$ catalyst with different morphologies.

**Table 4.** The characteristic temperature of CH$_4$ oxidation reaction of Pd/Y-ZrO$_2$ catalysts with different morphologies.

| Samples | $T_{50}$ (°C) | | | |
|---|---|---|---|---|
| | No Water | | With Water | |
| | F | A | F | A |
| Pd/FlYZr | 512 | – | – | – |
| Pd/SpYZr | 516 | 420 | – | 487 |
| Pd/ReYZr | – | – | – | – |
| Pd/BuYZr | 496 | 369 | – | 442 |

It can be seen from Figure 9b that the order of $CH_4$ oxidation activity of the aged catalyst is followed by Pd/BuYZr > Pd/SpYZr > Pd/FlYZr ≈ Pd/ReYZr. It is observed that the activity of Pd/FlYZr and Pd/ReYZr is completely deactivated, whether in a dry or wet atmosphere. On the contrary, the $T_{50}$ of aged Pd/BuYZr and Pd/SpYZr catalysts are 329 °C and 420 °C in a water-free atmosphere and 442 °C and 487 °C in a wet atmosphere, respectively. Of particular interest is that the aged Pd/BuYZr and Pd/SpYZr catalysts show higher methane oxidation activity than a fresh catalyst. Among them, Pd/BuYZr catalysts show the best methane oxidation activity. This may indicate that the particle size of Pd influences the performance of the catalyst. As shown in Table 2, the Pd particle size of fresh Pd/BuYZr and Pd/SpYZr catalysts was 2.0 and 2.3 nm. In comparison, after aging, they increase to 4.4 and 4.3 nm, respectively. It can be inferred that the relatively large Pd particles are more conducive to $CH_4$ oxidation. This observation is in agreement with the studies carried out by Stakheev [45] and Fujimoto [46]. Stakheev et al. [45] reported that the turnover frequencies (TOFs) of Pd/$\gamma$-$Al_2O_3$ increased with increasing Pd particle size in the range of 1–20 nm. The same trend has also been reported by Fujimoto et al. [46] for Pd/$ZrO_2$.

It should, however, be emphasized that the contribution of the crystalline phase of Y-$ZrO_2$ and the valence of active metal to the catalytic activity. As discussed in previous sectors, the Tetragonal phase of Y-$ZrO_2$ is more conducive to thermal stability. Compared with Pd/FlYZr and Pd/ReYZr catalysts, both Pd/BuYZr and Pd/SpYZr catalysts own better methane oxidation activity, which can be attributed to the stable single tetragonal phase structure of support. The XPS results discussed above clarify that metallic $Pd^0$ in the aged catalyst also affects the oxidation performance of $CH_4$. The presence of metallic $Pd^0$ is conducive to the formation of the PdO-Pd interface that promotes the dissociation of the C-H bond over $CH_4$ [47,48]. In general, the support material influences the Pd particle size and chemical state, which could play an important role in the methane oxidation reaction.

## 4. Conclusions

Y-modified $ZrO_2$ oxides with flower-like, spherical, reticulated, and bulk-specific morphology is prepared by hydrothermal synthesis. SEM, $N_2$ adsorption-desorption, XRD, CO pulse adsorption, $O_2$-TPO, TEM/HRTEM, XPS, and catalytic activity tests are conducted to systematically study morphology effect on the microstructure properties of the obtained Y-modified $ZrO_2$ oxides as well as the catalytic performance of their supported Pd/Y-$ZrO_2$ catalysts. It is found that the morphology affects the microstructures of Y-$ZrO_2$ and the chemical states of the active Pd species, thereby having a significant influence on the activity of methane oxidation. Among all the catalysts, bulk Pd/Y-$ZrO_2$ has the best catalyst performance and thermal stability. Y-$ZrO_2$ with block shape exposed the (101) surface, and the single tetragonal phase structure maintained after high-temperature aging is better for performance. The content of active $Pd^0$ species of Pd/Y-$ZrO_2$ with spherical and bulk morphology catalysts is significantly higher than that of Pd/Y-$ZrO_2$ with flower-like morphology catalysts, the relatively large-sized Pd particles and metallic $Pd^0$ jointly promote the catalytic oxidation of $CH_4$.

**Author Contributions:** Investigation, Writing—original draft, X.Z.; Conceptualization, Methodology and Wrting, T.Z.; Writing—review & editing, J.M. and C.W.; Resources, D.Y.; Supervision, P.N. All authors have read and agreed to the published version of the manuscript.

**Funding:** This research was funded by General Project of Yunnan Applied Basic Research Program (NO. 2019FB146) and the National Natural Science Foundation of China (NO. 22162015).

**Conflicts of Interest:** The authors declare no conflict of interest.

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
