# Peer review of "Morphology Effects on Structure-Activity Relationship of Pd/Y-ZrO2 Catalysts for Methane Oxidation"

_catalysts, doi:10.3390/catal12111420_

Round 1

Reviewer 1 Report

In this article, the authors prepared Y-ZrO2 with different morphologies using hydrothermal method and studied their methane oxidation activity. The results are interesting and backed up by experimental evidence. I recommend it for publication with minor revision.

1. In Figure 1 (a), Pd/BuYZr showed better performance than Pd/SpYZr with no water. However, with water, Pd/BuYZr showed worse performance than Pd/SpYZr. Can you give more explanation as to why the activity sequence has changed with the introduction of water?

2. I found the lines in Figure 1, specifically the conversion with water, a little hard to read because of the pattern of line. Maybe you could replot the figures with a different pattern?

3. In 3.2 section, line 246, the author stated that the diffraction peak was enhanced, suggesting that aging treatment only increases the crystallinity of the sample and did not change the phase structure. The intensity of the diffraction peak is usually not used to estimate crystallinity. However, if you could calculate the crystalline domain size using Scherrer equation, it might provide you with some more useful information. (Punctuation marks were not used correctly in this part, please double check)

4. In Figure 7 HRTEM, I found the red marking on the images hard to read. Can you make them more readable? Especially in (a2), I couldn't tell what it says.

Reviewer 2 Report

The article "Morphology Effects on Structure-Activity Relationship of Pd/Y-ZrO2 Catalysts for Methane Oxidation" is devoted to the study of the influence of the morphology of Pd/Y-ZrO2 catalysts obtained by hydrothermal and solvothermal synthesis on their activity in the methane oxidation reaction. In the article, the microstructure of the obtained materials is studied in detail by a complex of modern methods. However, corrections and additions must be made before publication.

1) The structure of the article is unusual, as a rule, the composition and structure of the received materials are described first, and then the activity.

2) in Figure 1, the dotted lines are poorly visible, it is necessary to improve

3) in page 6 line 205 - 82.662 m2 g-1 and 106.683 m2 g-1 values are specified with too much accuracy. What is the error of the BET method? Here and further, the area values must be brought into line. Similarly, in Table 2, data on area and Dp.

4) How was the particle size determined for materials with different morphologies in Figure 6? If this is the diameter for spherical nanoparticles, then there are several sizes for particles with complex morphology, it is necessary to explain.

5) There are practically no recent works from 2021-2022 in the list of references, it is necessary to add such references.

6) The methodology states that (Pd content was 0.5%), it is necessary to provide data on the actual composition of the catalysts obtained.

Round 2

Reviewer 2 Report

The authors have made the necessary additions and corrections. The article may be accepted for publication.